

# Machine Learning Techniques to Improve the Field Performance of Low-Cost Air Quality Sensors

Tony Bush[1,2], Nick Papaioannou[1], Felix Leach[1], Francis D. Pope[3], Ajit Singh[3], G. Neil Thomas[4], Brian Stacey[5] and Suzanne Bartington[4].

[1] Department of Engineering Science, University of Oxford, Parks Road, Oxford, OX1 3PJ, UK
[2] Apertum Consulting, Harwell, Oxfordshire, UK
[3] School of Geography, Earth and Environmental Sciences, University of Birmingham, Edgbaston, Birmingham, B15 2TT, UK
[4] Institute of Applied Health Research, University of Birmingham, Edgbaston, Birmingham, B15 2TT, UK
[5] Ricardo Energy and Environment, The Gemini Building, Fermi Avenue, Harwell, Didcot, OX11 0QR, UK

*Correspondence to*: Felix Leach (felix.leach@eng.ox.ac.uk)

**Abstract** Low-cost air quality sensors offer significant potential for enhancing urban air quality networks by providing higher spatio-temporal resolution data needed, for example, for evaluation of air quality interventions. However, these sensors present methodological and deployment challenges which have historically limited operational ability. These include variability in

performance characteristics and sensitivity to environmental conditions. In this work, field 'baselining' and interference correction using Random Forest regression methods for low-cost sensing of $NO_2$, $PM_{10}$, and $PM_{2.5}$ is investigated. Model performance is explored over 7 months obtained by field deployment alongside reference method instrumentation. Workflows and processes developed are shown to be effective in normalising variable sensor baseline offsets and reducing uncertainty in sensor response arising from environmental interferences. A mean absolute error of 2.5 ppb, 4.8 µg/m$^3$ and 2.9 µg/m$^3$ for $NO_2$,

$PM_{10}$, and $PM_{2.5}$ respectively, was achieved for corrected field-deployed sensors compared to a reference method. When used to correct data collected under environmental conditions outside model training, results meet European data quality objectives, albeit with lower accuracy than data from within the trained range.

## 1 Introduction

### 1.1 Air Quality Context

Poor air quality is recognised as the largest environmental risk to human health worldwide (Public Health England, 2018). Pollution levels in many UK cities regularly exceed legal limits and health-based guidelines and exert a national mortality burden equivalent to 28,000-36,000 deaths each year (Frank Kelly, 2018), with estimated economic costs of more than £20Bn. Road transport is widely recognised as the major urban air pollution source, particularly for $NO_2$ (Leach et al., 2020). Within this context in the UK, there has been continued policy commitment to tackling poor air quality through the UK Clean Air

Strategy (Defra, 2019; Defra and DfT, 2017). As a result, there is much demand for air quality evidence which can contribute to responsive decision making for pollutant mitigation interventions. In turn, low-cost sensor technologies have proved attractive, offering advantages over traditional instrumentation. These include lower operating costs (infrastructure,



commissioning and running costs), reduced administrative barriers (planning) and options for deployment in dense networks to deliver high spatio-temporal resolution datasets. One such setting which has adopted this approach is in the City of Oxford, where the 'OxAria' study has commissioned a low-cost sensor network to enhance regulatory grade air quality data for rapid assessment of COVID-19 related transport variations and local emissions control policy interventions including a proposed Zero Emissions Zone (National Institute for Health Research, 2020)

Low-cost, or at least more affordable air quality sensors, provide considerable potential to enhance spatial coverage of high-quality measurements which have historically been limited by the prohibitive cost of regulatory grade monitoring (Castell et al., 2017). Low-cost sensors offer potential for (i) a more agile and responsive technique for capturing the impact of air quality interventions and hotspots, being more flexible and quicker to deploy to capture the spatio-temporal variability in pollutant levels arising from specific emissions sources or influences of the built environment (Schneider et al., 2017), (ii) supplementing regulatory monitoring, modelling and source attribution evidence base for a better-informed population exposure estimates and policy decisions (Morawska et al., 2018) and (iii) opportunities for mobile air quality measurements and citizen science approaches that further challenge the traditional evidence base and democracy of information sources that contribute local air quality policy (Lim et al., 2019; Wang et al., 2021).

Low-cost sensors utilise and require (i) hardware which is both sensitive and specific to air pollutants at ambient levels; (ii) robust calibration and/or (iii) data processing methods to generate data of sufficient reliability and accuracy for the intended purpose(s) (Hasenfratz et al., 2012; Zimmerman et al., 2018) The latter present multiple methodological challenges: calibrations developed in the laboratory may not reflect real-world performance, resulting in sensor baseline drift and post-hoc data calibration is typically necessary to optimise data quality (Karagulian et al., 2019). For these reasons there remain concerns about data quality and reliability which imposes limitations upon current applications beyond a research setting (Bigi et al., 2018; Clements et al., 2019; Crilley et al., 2018, 2020; Woodall et al., 2017). However, their accelerated uptake in local authority settings is testament to their potential to deliver a new, high-resolution evidence base capable of contributing to modern policies for air quality management and public health protection.

## 1.2 Machine Learning Applications

Given the challenges and opportunities above, several studies have been undertaken using, primarily, machine learning (ML) algorithms, for low-cost sensor calibration and validation. ML techniques offer significant benefits in terms of utility over simpler methods such as multivariate regression and decision trees which can offer greater interpretive facility to understand and quantify the interfering factors. There is a trade-off, from an air quality domain perspective, between understanding and quantifying the sensor performance and developing satisfactory, practicable methods to support higher quality sensor





observations at the expense of knowing 'why and how much'. Given the setting for this research outlined above and more broadly, the current appetite for low-cost sensor data to support and influence local policy, data volumes and complexity of interferences, black-box ML approaches present greater utility. Techniques such as artificial neural networks (ANNs) (Esposito et al., 2016; Spinelle et al., 2017; De Vito et al., 2009), high-dimensional multi-response models (Cross et al., 2017), and multiple linear regression (MLR) models have been successfully used with variable results. In particular, ANNs have been

shown to be able to meet sufficiently low levels of uncertainty for certain pollutants such as ozone (Spinelle et al., 2017), but higher uncertainty levels for $NO_2$ persist.

Random Forest (RF) models present an alternative ML method which have shown promise as a tool for low-cost sensor calibration and validation. Zimmerman et al. used a RF regression model (RFR) for validation of co-located sensor for four

gases ($CO_2$, CO, $O_3$ and $NO_2$) and found error rates of <5% for $CO_2$, ~10–15% for CO and $O_3$, and 30% for $NO_2$. These estimates were within the precision and accuracy error metrics from the US EPA Air Sensor Guidebook for personal exposure (Tier IV) monitoring (Zimmerman et al., 2018).

RFs are an ensemble decision tree approach which employ multiple decision trees to solve regression and classification

problems. They are a bagging technique, growing their decision trees in a bootstrap fashion (random sampling with replacement). A final prediction of the target value (in our case the reference method air quality concentration) being made as an aggregation (average) of the values estimated by the component trees.

Decision trees are known to be prone to overfitting, especially when allowed to grow deep, because after bootstrap sampling,

their trees are grown by considering all sampled features at each decision node. RFs use an alternative, improved tree growth method which tends to limit this propensity for overfitting. The RF method achieves this by adding greater diversity to the data used to train its decision trees. As a result, predictions from all trees have less correlation and, therefore, when aggregated a better prediction. RFs do this by selecting a random subset of training features for consideration at each decision node for each bootstrapped sample. Consequently, even if by chance, the same bootstrapped sample were selected to train two trees,

the resulting trees will likely to be different because subsequent random sampling of features at each decision node (Breiman, 1996).

A generic example of a two variable regression problem is presented in Fig. 1. In this figure, the decision tree (on the left) splits the parameter space into partitions (branches) based on logical operators on criteria relating to the parameter space

(variable X* < 0.* etc.). These operations continue until a terminal node is reached. At this point, a single prediction is made which is the average of all the available values that the dependent variable takes in that partition. The same process is navigated for more than two features, however the parameter space becomes non-trivial to visualise.



One major problem that decision trees can suffer from is high variance (Hastie et al., 2009). Often a small change in the data
can result in a very different series of splits and to a large change in the structure of the optimal decision tree. At least in part,
this specificity of decision trees contributes to a tendency to overfit which results in models that do not generalize well to
unseen data / situations. Although methods to manage this behaviour exist, they add an extra burden and are either not needed
by RF models or included out-of-the-box.

The disinclination of RF models to over-fit is a key advantage of the technique and comes from the bagging and random feature
selection methods employed. They build a diverse ensemble of many weakly correlated predictors (decision trees) which, at
run time, predict based on the modal class (in classification models) or the average of all predictions (regression models). It is
the diversity of predictions and their prediction error that present advantages for RF models, as when averaged to make the
ensemble prediction, they often result in better performance than decision trees.


From an operations perspective they offer benefits to the multivariate regression problems presented in this paper being; (i)
tolerant of multiple collinearity, which is intrinsic to the air quality datasets of interest; (ii) suffer less from over-fitting and
therefore promote a well generalised model which is adept to deployment across multiple datasets derived from different sensor
locations; (iii) do not require data transformation for optimisation, thereby simplifying the data logistics and computational
burden; (iv) handle multiple inputs variables with ease; (v) relatively easy to deploy, train and test across common desktop
computer environments available to air quality practitioners.

This study further develops practicable methods for enhancing low-cost air quality sensor data uncertainty. Whilst ML
techniques are established for low-cost air quality sensor validation with co-located sensors for $NO_2$ (and other gases), in this
study we aim to advance the base-lining strategies of low-cost air quality sensors by repurposing existing analytical techniques
which, to the best of our knowledge have not previously been used for field baselining and interference correction. In addition,
we apply RF algorithms to low-cost particle sensors. We present an approach which utilises an RFR to predict and compensate
for interferences from multiple environmental parameters upon the sensor signals. These methods offer a flexible, extendable,
and reusable technique(s) to account for drift/changes in sensor calibration that can commonly occur in the field, in addition
to a correction model to compensate for environmental interferences from, for example, temperature and relative humidity
amongst others.

## 2 Methods and Materials

### 2.1 Air Quality Instrumentation

The sensor technology used in this research was the Praxis Urban sensor system supplied by South Coast Science Ltd. The
units were equipped with an Alphasense NO2-A43F electrochemical $NO_2$ sensor (Alphasense Ltd., 2019a) and an Alphasense





N3 optical particle counter (OPC) (Alphasense Ltd., 2019b). The sensor system sample rate was set to 10 second intervals. Reference measurements of ambient $NO_2$, $PM_{10}$ and $PM_{2.5}$ were obtained from the Defra, Oxford St Ebbe's, Automatic Urban & Rural Network (AURN) monitoring station (UKA00518) (Defra, 2021). The St Ebbe's monitoring is located in South Oxford in a residential area, approximately 250 m from the nearest main road, as such it presents a classic urban background

environment. St Ebbe's employs a Teledyne T200 chemiluminescence $NO_x$ analyser and a Palas FIDAS 200 fine dust aerosol optical spectrometer. Both the Praxis Urban sensors and the AURN sensor inlets are located at a height of 2.7m and 8m from the nearest minor road. The reference methods are designated type approved reference instrumentation for regulatory compliance monitoring (Defra, 2013). Reference measurements were obtained at 15-minute average resolution by special arrangement with the network operators for the period 1st June to 31st December 2020. Official 1-hour time resolution datasets

were considered too coarse for RF model development and sourcing of higher time resolution data was, therefore, essential for the characterisation of the transient interferences.

### 2.2 Air Quality Datasets

The OxAria project maintains a network of 16 operational Praxis units within the City of Oxford. Measurements obtained from

the OxAria sensor, co-located at the Oxford St Ebbe's AURN monitoring station was the primary source of data for model development as part of this work. Sensor data were aggregated to a 15-minute average resolution from the initial logging interval of 10s to ensure conformity with the datum for the AURN datasets. The quality assurance status of the AURN datasets was valid / verified.

### 2.3 Sensor Baseline Offset Correction

The rationale for the baseline correction was to prepare sensor datasets for interference correction using an RF model. There was evidence for variability in the baseline OxAria $NO_2$ sensors (more details are in the results section), however, there was no obvious baseline variability in the PM data. Any variation in the baseline conditions at a network level will confound comparisons undertaken across the network and with air quality limit values and guidelines, irrespective of the pollutant

species. Importantly, baseline variability will also be problematic for the deployment of a generalised RF correction model, the characteristics of which will be 'locked-in' to the baseline of the dataset used for its training; in this case, the collocated sensor at St Ebbes which has a baseline offset of approximately +80 ppb $NO_2$. To address this issue, a series of filters and baseline identification techniques were developed to adjust for variance in sensor signal and correct for the sensor baseline in a systematic and automatable way. This method normalises the sensor baseline(s) across the OxAria network and has been

applied to the $NO_2$, $PM_{10}$ and $PM_{2.5}$ datasets. The 4-stage processing approach is summarised in the schematic presented in Fig. 2 and is outlined in more detail in the sections below.



### 2.3.1 Stage 1 – Empirical Filters for Removal of Outliers and Anomalies.

The filtering criteria presented in Table 1 were identified empirically from an analysis of typical sensor performance from the sensor network and from similar parameters logged at the St Ebbe's AURN station. Sensor observations falling outside of these bands of acceptance were excluded from further analysis. Filters for $NO_2$ and particles are presented in Table 1. Filters (i) and (iii) removed data points outside of conservative estimates of the normal range of ambient temperature in Oxford thereby excluding any anomalies arising from temperature dependent sensor system corrections that may be performing out of range. Filters (ii) and (iv) performed a similar role for relative humidity. Filter (v) removed particles data during periods of low OPC sample flow rate.

### 2.3.2 Stage 2 - Baseline Identification & Offset / Drift Correction.

Stage 2 implemented a statistical method developed in the analytical domain for baseline correction in chromatography and Raman spectroscopy. Adaptive Iteratively Reweighted Penalised Linear Squares regression (airPLS) (Z.-M. Zhang, 2011; Zhang et al., 2010) was used to systematically correct sensor datasets after stage 1 filters had been applied. Performance and flexibility were a key factor in selection of a preferred method for baseline correction. airPLS does not require significant user intervention to perform satisfactorily, nor prior information or supervision, e.g. peak detection. It is a fast, flexible technique, and readily deployable in code.

### 2.3.3 Stage 3 – Baseline Over-fit Compensation

airPLS is highly efficient in correcting a baseline to zero, an artefact that derives from its intended application domain (chromatography) where a zero baseline is generally encouraged. Stage 3 applies a compensation method for the efficacy of the airPLS algorithm in correcting sensor baseline to zero, which in effect removes the urban, regional, and rural background contributions from the sensor signal. The method scales Stage 2 outputs by the difference between the baseline identified in the Stage 2 corrections and the baseline of the city scale background, (observations from Oxford St Ebbe's, urban background AURN station). For $NO_2$ this compensation method resulted in an average uplift of +2.4 ppb. For $PM_{10}$ and $PM_{2.5}$ the uplift was +2.6 and +1.5 µg/m$^3$ respectively.

### 2.3.4 Stage 4 – Residual Error Removal

The final stage of the correction method accounts for any remaining residual anomalies that present as negative concentrations not accounted or corrected for in stages 1-3. The impact of this stage on the sample population was intended to be negligible. Taking $NO_2$, for example, stage 4 processing identified only 0.4% of the sample population as negative values and removed them from downstream RF modelling work.



### 2.4 Sensor Interference Correction with Random Forest Regression Modelling

The following sections present the configuration of the RF model and approach to training of the regression model. The RF modelling was carried out in Python implemented using the SciKit-Learn open-source machine learning library (Pedregosa et al., 2011).

#### 2.4.1 Feature Engineering

Feature engineering describes the process of creating new training features (variables) that are more illustrative of the underlying problem being modelled. The aim of feature engineering is to affect better model training and performance. It is a common pre-processing step in RF modelling and many other regression and classification techniques (Breiman, 2001; Yu et al., 2011).

Feature engineering was constrained in scope and complexity by the need to deploy the model across a network of sensors. Hence, feature datasets must be readily available or replicable throughout the network of sensors. Table 2 presents the features used in model training of the pollutant specific correction model. The source of the training feature is presented in the 'type' column.

#### 2.4.2 Random Forest Regression Model Training

RF model training was performed with co-located sensor and reference measurements acquired at the St Ebbe's AURN monitoring station over the period June to November 2020. After feature engineering (above), the core dataset was split into training and validation datasets using a 75% and 25% split, respectively. This 'hold-out' validation method was combined with a K-Fold cross-validation approach (Berrar, 2018) to estimate the performance of the model in terms of the mean absolute error score (MAE).

In many cases, RF models work reasonably well with the default values for the hyper-parameters specified in the software packages (Probst et al., 2019). Even so, for standardisation across pollutant applications and computational efficiency we considered constraining the models using tree size metrics – number of trees, maximum number of leaf nodes and the minimum number of samples required to split an internal node.

The maximum number of leaf nodes hyper-parameter was established by way of a cross validation sensitivity test on an array of 10 to 5,000 nodes (node spacing set to 50). The cross-validation exercise fitted an RFR model to the input feature dataset and iterated over the array of nodes to predict the MAE. Cross validation results for $NO_2$ are presented in Fig. 3. These Are illustrative of similar behaviours for $PM_{10}$ and $PM_{2.5}$. Figure 3 shows the MAE decreasing as a function of increasing maximum number of leaf nodes (model complexity). Cross validation results similar to those presented in Fig. 3 were used to identify



the optimum number of leaf nodes for each pollutant-specific model - the point on the x-axis where increased model complexity delivers only marginal improvement in MAE for training, validation and cross validation test samples. The process was repeated for the $PM_{10}$ and $PM_{2.5}$ models. Figure 3 also confirms some assumptions about RFR model training in general:

- Gains in MAE quickly drop-off with increasing feature numbers,
- For RF model predictions which are based on an ensemble average of all trees, the MAE of predictions based on training data will tend towards but never reach zero,
- K-fold cross validation produced the most conservative estimates of model accuracy (highest MAE).

The maximum of 3,500 of leaf nodes was established by this cross-validation process for the $NO_2$ RFR model whereas the same hyper-parameter for both $PM_{10}$ and $PM_{2.5}$ models was set at 3,000 nodes. The minimum number of samples allowed in a single partition was set to 2.

Having established the maximum number of leaf nodes for the three pollutant-specific models ($NO_2$, $PM_{10}$ and $PM_{2.5}$), the
235 number of trees was determined. Best-practice on setting the optimum number of trees within RF is variable with advice ranging from between 64-128 (Oshiro et al., 2012) For this research, the incremental improvement in MAE arising from between 100 and 500 trees was evaluated. Results did not show significant improvement in model MAE over this range within the context of typical ambient air quality concentrations expected. The number of trees used was set to 100 to minimise computational cost during training. Table 3 presents a summary of the hyperparameters used in the training of each Random
Forest model.

## 3 Results and Discussion

### 3.1 Uncorrected sensor data

Figure 4 presents the 3-hour rolling mean of 'raw' real-world $NO_2$ observations from three OxAria low-cost electrochemical sensors and a reference method i.e. sensor data outputs before any correction algorithms are applied. The rolling 3-hour mean
is presented to attenuate noise in the datasets for visualisation. Sensor A and the reference method are co-located at an urban background location, Sensor B is located at an urban centre location, and Sensor C at a roadside location. The sensor systems are identical and were calibrated at the same time in the same laboratory. Figure 4 shows a comparatively low signal to noise ratio in the sensor's observations when compared with the reference method and marked variability in the baseline(s) which confound interpretation of the pollutant levels. The severity of the variability in sensor baseline offset is further contextualised
when sensor location is considered (as noted above). Sensor A being at the urban background is far from significant $NO_2$ emissions sources, whereas Sensors B (urban centre) and C (roadside) are comparatively close to major road transport emission sources. Despite their relative proximity to emission sources the baseline for the urban background sensor is ~40 ppb higher




than its urban centre / roadside neighbours. Given that the sensors were calibrated to the same standard within a laboratory environment prior to deployment in the field, our assumption is that the sensor baselines have been influenced in some way after calibration, then stabilised as shown. In addition, frequent spikes in the sensor trace(s) can be observed which manifest as both short lived, transient events of ~10 s duration in the 100-500 ppb range and as longer-lived 60 s+ events, frequently in the 1000-2000 ppb range. This sort of sensor behaviour is linked to multiple environmental interferences of which temperature and relative humidity are amongst the most important (Spinelle et al., 2015). We anticipate that these sensor characteristics are replicated across the OxAria sensor network and replicated throughout similar sensor networks using electrochemical $NO_2$ sensors.

### 3.1 Sensor Baseline Offset Correction Results

Figure 5 presents the incremental outputs of each stage of the sensor baseline correction model described in section 2.3. As an example, co-located $NO_2$ sensor and reference method observations from St Ebbe's are presented for August 2020. This sensor and fragment of the 2020 time series was chosen as illustrative of the performance of the model on a sensor of known offset (~80 ppb) and the general effect of each stage in the correction process.

Commenting individually on each stage presented in Fig. 5; Fig.5a indicates the presence of a clear offset in the $NO_2$ sensor signal of ~+80 ppb relative to the co-located reference method. Fig.5b presents the outcome of applying empirical filters to screen out anomalous sensor behaviours and data outliers. Noticeably for this location, the empirical filters have screened out observations around 10 August but left the 250+ ppb spike in concentrations on 13 August in place. Fig.4c presents the removal of the sensor baseline using airPLS and Fig.5d compensation for its efficacy; the baseline of the corrected sensor time series and reference method baseline are recalculated (again using airPLS) and the sensor baseline scaled by the difference in the two terms. The last step shown in Fig.5e removes any residual negative errors not already captured.

The data presented in Fig. 5 and other locations in the OxAria network have shown the airPLS based baseline correction model to be effective at normalising a variable baseline shown in the $NO_2$, $PM_{10}$ and $PM_{2.5}$ sensor signals across the network. The method also maintains the fidelity of the dynamic range of the original sensor signal. Its effectiveness facilitates the training of generalised RF correction models. In terms of optimisations, the approach was relatively insensitive to changes in the configuration of the empirical filters applied in stage 1 corrections and the lambda value of the airPLS technique which controls the order of smoothing applied to the baseline estimate.

The over-fitting of the corrected sensor baseline (to zero ppb) introduced by the efficacy of the airPLS technique is compensated for by rescaling of the sensor baseline to that of the city background. If this is an over-simplification of the experimental error handling it is a reasonable trade-off given the volumes of data involved and computational logistics involved overall.



The availability of a reliable and high-quality city background at a time resolution comparable to that of sensor observations e.g. at most 15-minutes averages, is essential for effective screening transient anomalous sensor behaviours which skew sensor datasets significantly and mask important underlying data structure or anomalies. We also note that reference method data

resolved to these time resolutions is difficult to obtain in the UK and acknowledge the support of Oxford City Council and Defra's AURN QA/QC management team in securing these data for this study.

## 3.2 Random Forest Correction Modelling Results

### 3.2.1 Random Forest Regression Model Training

Outputs from the model training exercise are shown in Fig.s 6-8 as a series of regression plots for the RFR models developed

for $NO_2$, $PM_{10}$ and $PM_{2.5}$. For each pollutant, three regression plots are presented to illustrate (i) the relationship between the baseline corrected sensor observations and reference method (left), (ii) the same relationship constrained to the validation subset (middle) and (iii) the relationship between the corrected sensor observations and reference method. A simple ordinary least squares (OLS) regression analysis is presented in each case to describe each relationship.

The plots to the right of Figs. 6-8 show that the respective RF models are highly effective in predicting the target observations (reference method). In doing so, they demonstrate their capability to predict the combined interferences from a variety of environmental factors found in the data of the left and middle regression plots. The left and middle plots also show that training and validation datasets come from the same sample population (one having been randomly sampled from the other) providing a useful internal validation of model training to reflect variations in training features. Further checks on the models using

unseen data from outside of the sample populations will better test likely performance of the models in the field.

Figs. 6-8 show the dramatic impact of the RF model correction as demonstrated by the coefficients of variation in each of the three cases. The R-squared value of corrected sensor vs reference method observations is a convenient evaluator for the ability of the models to capture the variability in the dependent datasets. Clearly, the $PM_{2.5}$ model performs excellently in this respect

with an R-squared value of 0.96 and OLS slope and intercept terms approaching unity. The respective R-squared value for both $PM_{10}$ and $NO_2$ RF models (0.84 and 0.86) also indicate good model performance. The values for R-squared above are consistent with the out of bag scores achieved at training time (0.85, 0.82 and 0.91 for $NO_2$, $PM_{10}$, and $PM_{2.5}$ respectively) which provide an additional check on model performance using data not explicitly used in the training. Even so, it is clear from Figs. 6 and 7 that the models struggled, on occasion, to accurately predict higher reference concentrations and $NO_2$ and

$PM_{10}$ predicted concentrations are generally more scattered compared with $PM_{2.5}$. It is also noticeable that in all three cases the RF models are biased, tending to under-predict the reference concentration as demonstrated by the regression equation slope terms and this is particularly noticeable in the 15 ppb+ concentration unit range.





### 3.2.2 RF Correction Performance Characteristics (hold-out validation set)

The performance of each component of the correction method is presented in Table 3 in terms of the MAE delivered by
correction outputs at each stage, relative to the reference method observations. Table 3 shows that the RFR correction adds
significant value to the baseline correction alone contributing to a further 90-95% reduction in the MAE terms. In concentration
units this equates to fully corrected $NO_2$ sensor observations within approximately ±1 ppb of the reference observation. Similar
comparisons for $PM_{10}$ and $PM_{2.5}$ indicate corrected concentrations within ±1-2 µg/m³ of the reference method. These compare
favourably with results in the literature for all three pollutants.


The impact of corrections to this order of magnitude upon the sensor time series can be visualised in Figs. 9-11 which presents
the uncorrected-baseline normalised sensor observations, fully corrected sensor observations and reference observations for
$NO_2$, $PM_{10}$ and $PM_{2.5}$. Figure 9 shows that for $NO_2$ there is some visual evidence of the RFR model over correction (relative
to the reference method) during periods of peak concentration, particularly in mid to late June and August. Otherwise, the $NO_2$
correction tracks that of the reference observations well.

### 3.2.3 RF Correction Model Performance Characteristics (unseen data)

Table 4 presents estimates of the performance of the correction models based on unseen data from December 2020 i.e. data
not previously used for model training nor validation. Table 4 shows, as expected, that model performance for unseen data is
less favourable compared with the validation data, returning higher values for the MAE metric, particularly for the $PM_{10}$ model.

In late November / December 2020 and latterly, continuing through quarter one of 2021 (not shown), the sensor network
observed episodes of high particle concentrations which coincided with a drop in ambient temperature (and dew point
temperature) to the order of 10°C. Reciprocal changes in relative humidity are not observed, nor is there an obvious change in
sensor sample flow rate. It is also notable that similar conditions were not commonplace throughout the model training dataset
(June to November 2020). The episode conditions observed by the sensor network are not replicated in the reference method
dataset and are likely the main driver for the increase in the MAE for the particulate matter correction models shown in Table
4.

Despite these issues, the models deliver substantial improvements on raw dataset (not shown) and baseline-adjusted data
(shown), with relative improvements in sensor error of 37-94% in the latter; equivalent to corrected observation within, on
average, approximately ±3 ppb of the reference method for $NO_2$, ±5 µg/m³ for $PM_{10}$ and ±3 µg/m³ for $PM_{2.5}$.



The decrease in model performance when targeted at unseen data and the observations on ambient conditions and sensor
operation above, illustrate the need for long time series for model training, and certainly time series that cover all environmental
conditions to which the sensors will be exposed.

The MAE value for $PM_{10}$ and to lesser extent that of $PM_{2.5}$ in December indicates that these pollutants displayed sizeable
variance during this period. This can be observed in the time series for December 2020 presented in Figs. 12-14 which shows
that both $PM_{10}$ (Fig. 13) and $PM_{2.5}$ (Fig. 14) exhibited a number of large peaks in concentrations or episode events. Despite
these events in the unseen data, the correction model outputs demonstrate that they generalise sufficiently to remove most of
the interference(s).

**3.2.3 Corrected Sensor Performance vs. European Air Quality Data Quality Objectives**

European Ambient Air Quality Directives (European Commission, 2004, 2008) have established data quality objectives (DQO)
which must be met to perform specific types of regulatory measurement tasks. These DQOs include, amongst other criteria, a
minimum requirement for the expanded uncertainty of measurements. Under these regulations, 'indicative' assessment
methods, those that can be used to supplement reference and / or equivalent methods, require an expanded uncertainty estimate
of ±25% and ±50% for $NO_2$ and particles measurement methods, respectively. These criteria are important given that indicative
assessment is the most likely niche for low-cost sensors within the regulatory assessment tool kit. Comprehensive guidance on
the calculation of expanded uncertainty has been provided by the European Commission Working Group on Guidance for the
Demonstration of Equivalence (EC Working Group, 2010, 2020) in addition to a convenient spreadsheet tool to support
traceable calculation of appropriate metrics .

Table 5 presents the expanded uncertainty estimates for corrected sensor observations. These estimates were calculated using
the spreadsheet tool (EC Working Group, 2020) to provide a further performance indicator on the adequacy of the corrected
sensor data for air quality assessment applications. From Table 5 we see that the corrected sensor outputs for all pollutants
perform well relative to the target expanded uncertainty criteria recommended by European legislation. The expanded
uncertainty estimates for the corrected sensor data from the validation dataset, (data not used in RF model training), are within
the prescribed limits, 21%, 34% and 18% respectively for $NO_2$, $PM_{10}$ and $PM_{2.5}$. In addition, guidance on the calculation of
expanded uncertainty (EC Working Group, 2010), allows for the correction of slope and intercept terms in the relationship
between sensor and reference method. The result for each model is also presented in Table 5 and demonstrate further reductions
in the expanded uncertainty estimates can be achieved, to levels that approach those of the reference method itself (Defra,
2013) and certainly within the equivalence thresholds (±25%) established by the European Commission Working Group on
Guidance for the Demonstration of Equivalence. In tests based on the validation set, expanded uncertainty estimates for RFR
model corrected observations for $NO_2$, $PM_{10}$ and $PM_{2.5}$ were 4%, 12% and 10% respectively. Highly respectable coefficients
of determination are achieved between reference and corrected sensor observation in all cases.



Results from reciprocal tests based on unseen data are presented in Table 6 using corrected sensor and reference method observations from December 2020. Comparing Tables 5 and 6, an increase in uncertainty is observed. This is expected, given
that statistics presented in Table 5 are based on the *validation* dataset, taken at random, from the same sample population as that used for model training, whereas Table 6 is based on entirely *unseen* data. Despite observed increases in uncertainty, corrected sensor observations continue to perform well relative to their target DQOs. All corrected datasets meet the criteria recommended by European legislation and the corrected expanded uncertainty estimate for $NO_2$ (10%) is particularly noteworthy.


Also evident from Tables 5 and 6 is the decrease in the performance of the $PM_{10}$ and $PM_{2.5}$ correction models on the unseen dataset. December 2020 saw the occurrence of several pollution events in the particle sensor time series (as also noted above). Although these events were observed throughout Oxford in multiple particle sensor time series, they were not reciprocated in reference measurements, nor in $NO_2$ data. We theorise, therefore, that they are an artefact of local meteorological events
interfering with the OPC sensor operation. Occurrence of similar events in training data is limited and offers (i) an explanation of comparative underperformance of $PM_{10}$ and $PM_{2.5}$ correction models during December 2020 dataset and (ii) an illustration of the importance thorough training of the correction models using long-running datasets which provide adequate coverage of likely environmental interferences and their anticipated variation.

## 4 Conclusions

This study has demonstrated a simple and effective method for attenuating the effects of sensor baseline variability and interferences from ambient environmental parameters upon low-cost electrochemical and optical particle counter sensor signals.

The methods presented in this paper have been tested at a high temporal resolution against co-located reference method
observations from the UK's regulatory monitoring network (AURN). Using MAE as an indicator of sensor error (relative to reference observations), the methods developed can reduce the error in $NO_2$, $PM_{10}$ and $PM_{2.5}$ observations from the low-cost sensors tested by 90-95% (based on model validation data not used in RF training). In the case of the low-cost $NO_2$ sensor, corrections reduced the MAE of sensor observations to within ± 1 ppb of the reference observation. Similarly, for $PM_{10}$ and $PM_{2.5}$ MAE estimates were within ± 2 $\mu g/m^3$ and ± 1 $\mu g/m^3$ respectively. The R-squared value achieved for corrected $NO_2$ and
$PM_{10}$ sensor observations was 0.84 and 0.92 for $PM_{2.5}$.

Tests on how the methods generalised to unseen conditions have shown that RFR correction models, trained on 6-months of data from June to November 2020, are tolerant of a wide range of competing environmental interferences. Tests based on



unseen data from December 2020 delivered MAE estimates for corrected low-cost $NO_2$, $PM_{10}$ and $PM_{2.5}$ sensors ±3 ppb, ±5 µg/m$^3$ and ±3 µg/m$^3$ respectively.

Given these indicators for the level of improved uncertainty that can be achievable with low-cost sensors, we propose that data from reputable, quality devices can now play a meaningful role in the air quality assessment toolkit. Indeed, using the methods presented, sensor data are likely to deliver at least comparable data quality to passive sampler methods (for $NO_2$), with the benefit of higher temporal resolutions.

To substantiate this observation, this paper has presented data demonstrating that the RF based methods are capable of delivering corrected low-cost sensor data that meet the general requirements for 'indicative measurements' as set out by the European Ambient Air Quality Directive. In doing so, we have used methods prescribed by the European Commission Working Group on Guidance for the Demonstration of Equivalence to calculate expanded uncertainty estimates for corrected sensor observations. For tests based on both validation and unseen datasets, the expanded uncertainty of corrected sensor data was within the requirements set by the European Ambient Air Quality Directive for indicative monitoring (within ±25% of the reference observation for $NO_2$, ±50% for particles). Indeed, these tests showed that the corrected expanded uncertainty estimates were within or proximal to the equivalence thresholds (±25%) established by the European Commission Working Group on Guidance for the Demonstration of Equivalence. In tests using unseen data, the most stringent test available to the study, the expanded uncertainty estimates for RFR model corrected observations for $NO_2$, $PM_{10}$ and $PM_{2.5}$ were 10%, 24% and 29% respectively.

Demonstrating conformance with these regulatory thresholds in a traceable way is a significant milestone, not least for the potential to unlock applications as 'supplementary assessment' method for compliance assessments but also within the context of the stringency of the acceptance criteria, and the rigour of the expanded uncertainty calculation method set out by the Working Group.

**Funding**

This research was funded by The Natural Environment Research Council, grant number NE/V010360/1. Its forerunning pilot project was funded by The National Institute for Health Research, grant number NIHR130095. This publication arises in part from research funded by Research England's Strategic Priorities Fund (SPF) QR allocation.



**Acknowledgments**

The authors would like to extend thanks and gratitude to the Chair of the project Steering Committee, the late Prof Martin Williams, for his guidance and encouragement over the years. The authors would also like to thank the study steering group
for their technical contributions and administrative support and Ricardo Energy & Environment in facilitating data acquisition. Oxford City Council and Oxfordshire County Council are both thanked for advice and support.

**Author contributions**

FL, NP, SB and TB conceived the study. TB and FL corralled the data. TB and NP performed data analysis. BS supplied reference observations at the required resolutions and finessed expanded uncertainty estimates in accordance with the guidance.
TB drafted the paper. All co-authors contributed to reviewing and editing the paper.

**Conflicts of Interest**

The authors declare no conflict of interest.



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





**Figure 1: Visual representation of a generic, two variable Decision Tree regression problem (left) and its mapping on to a parameter space for the independent variables (right).**


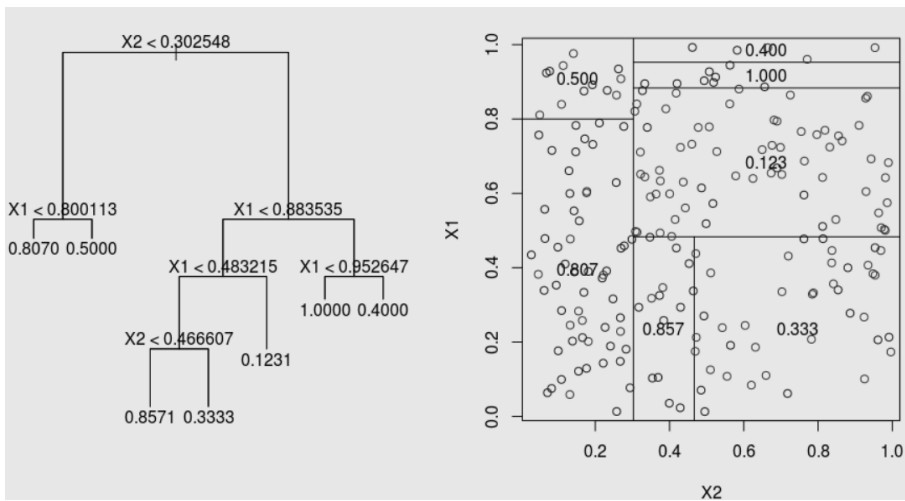

**Figure 2. Schematic of the sensor baseline correction model including interfaces with downstream RFR interference correction model.**

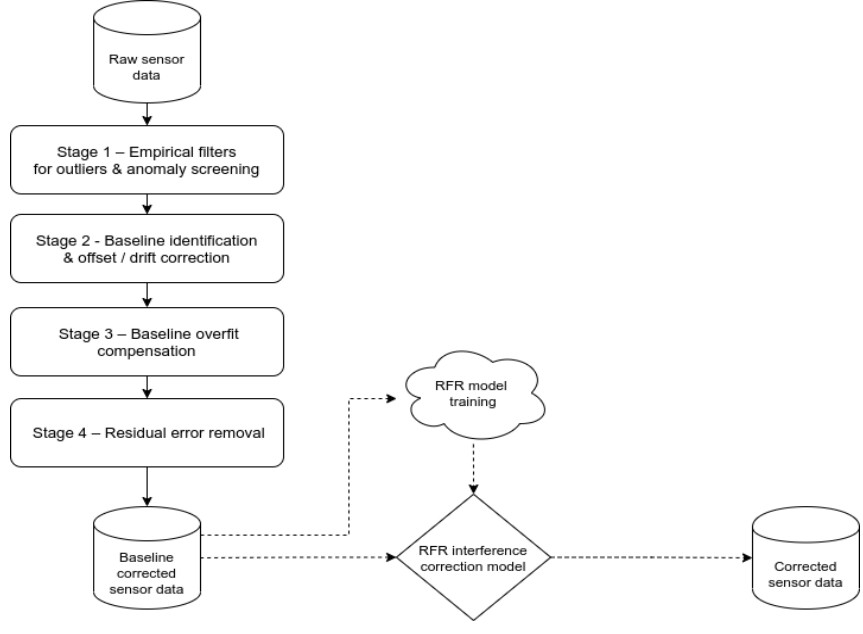




**Table 1. Empirical filters used for screening out anomalous sensor data.**

| Empirical filters - NO$_2$ | | Empirical filters – PM$_{10}$ and PM$_{2.5}$ | |
|---|---|---|---|
| (i) | -10 °C < temperature < 35 °C | (iii) | -10 °C < temperature < 35 °C |
| (ii) | Relative humidity > 35% | (iv) | Relative humidity > 35% |
| | | (v) | Sample flow rate > 2 ml/min |

**Table 2. Model feature (variables) used in RF model training and prediction by pollutant model.**

| Model | NO$_2$ | PM$_{10}$ | PM$_{2.5}$ | Type |
|---|---|---|---|---|
| Sensed concentration / mass | √ | √ | √ | Stock |
| Working electrode voltage | √ | x | x | Stock |
| Auxiliary electrode voltage | √ | x | x | Stock |
| Corrected working electrode voltage (offset corrected) | √ | x | x | Stock |
| Sample flow rate | x | √ | √ | Stock |
| Sample time of flight | x | √ | √ | Stock |
| Temperature | √ | √ | √ | Stock |
| Relative humidity (RH) | √ | √ | √ | Stock |
| Rate of change in temperature at T-15 mins | √ | √ | √ | Engineered |
| Rate of change in temperature at T-30 mins | √ | √ | √ | Engineered |
| Rate of change in RH at T-15 mins | √ | √ | √ | Engineered |
| Rate of change in RH at T-30 mins | √ | √ | √ | Engineered |
| Hour of day | √ | √ | √ | Engineered |
| Day of week | √ | √ | √ | Engineered |
| Rush hour classifier | √ | √ | √ | Engineered |

'Stock' indicates a feature based directly upon logged sensor observations, 'Engineered' indicates a featured

derived from re-analysis of one of more stock features.





**Figure 3. NO₂ RFR model performance returns with increasing model complexity (maximum number of leaf nodes included in training, validation and cross-validation datasets).**

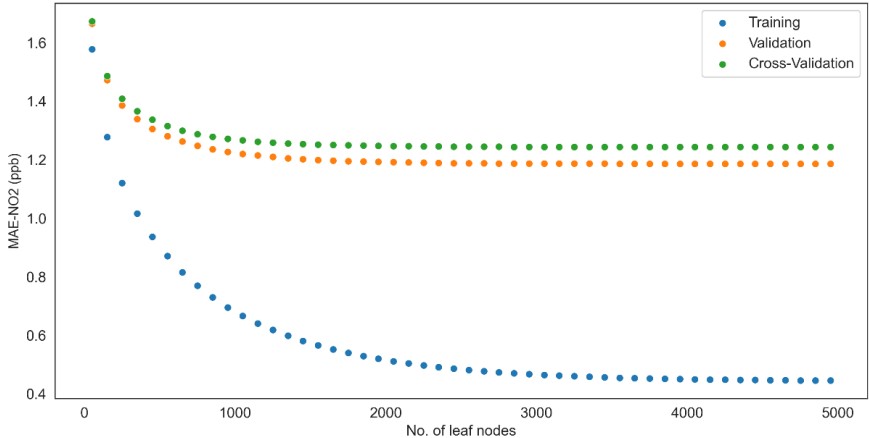

**Table 3. Summary of Random Forest hyperparameter setting used in model training.**

| Hyperparameter | Model Type | | |
| --- | --- | --- | --- |
| | NO₂ | PM₁₀ | PM₂.₅ |
| No. of trees | 100 | 100 | 100 |
| Criterion | 0 | 0 | 0 |
| Max. tree depth | 0 | 0 | 0 |
| Min. samples per leaf node | 1 | 1 | 1 |
| Max. no. of leaf nodes | 3500 | 3000 | 3000 |
| Min. sample per node | 2 | 2 | 2 |
| Min. leaf node weight fraction | 0 | 0 | 0 |
| Min. impurity decrement | 0 | 0 | 0 |
| Min impurity split | 0 | 0 | 0 |
| Max. no. features | 15 | 15 | 15 |
| No. jobs | -1 | -1 | -1 |
| Bootstrap sampling | 1 | 1 | 1 |




**Figure 4. Three hour rolling mean raw low-cost sensor and reference method NO₂ time series at three locations in Oxford 2020 (The y-axis is discontinuous to allow structure in data below 200 ppb and upper extrema to be displayed).**

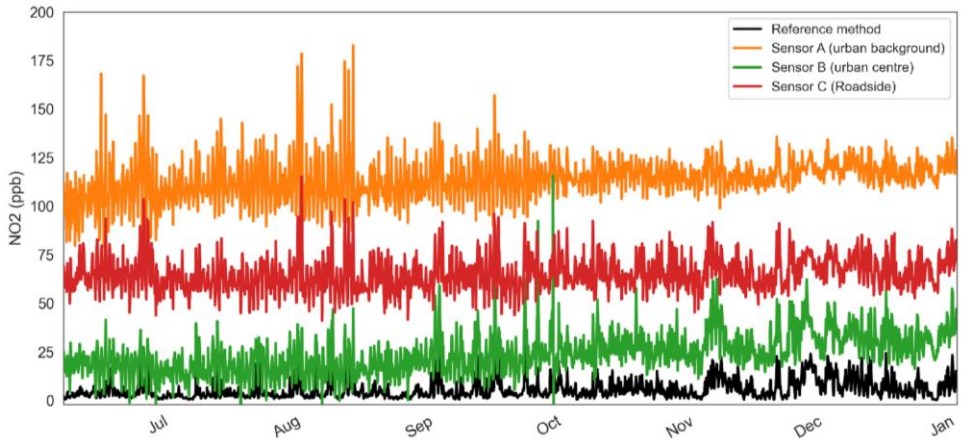






**Figure 5. (a-e) Illustrative impacts of each stage in the sensor baseline offset correction model, St Ebbe's, August 2020.**

Fig.5a - Raw sensor signal & reference method

Fig.5b – Correction 1. Application of empirical filters for anomaly & outlier removal

Fig.5c – Correction 2. Baseline offset correction

Fig.5d – Correction 3. Compensation for efficacy of baseline offset correction

Fig.5e – Correction 4. Removal of residuals

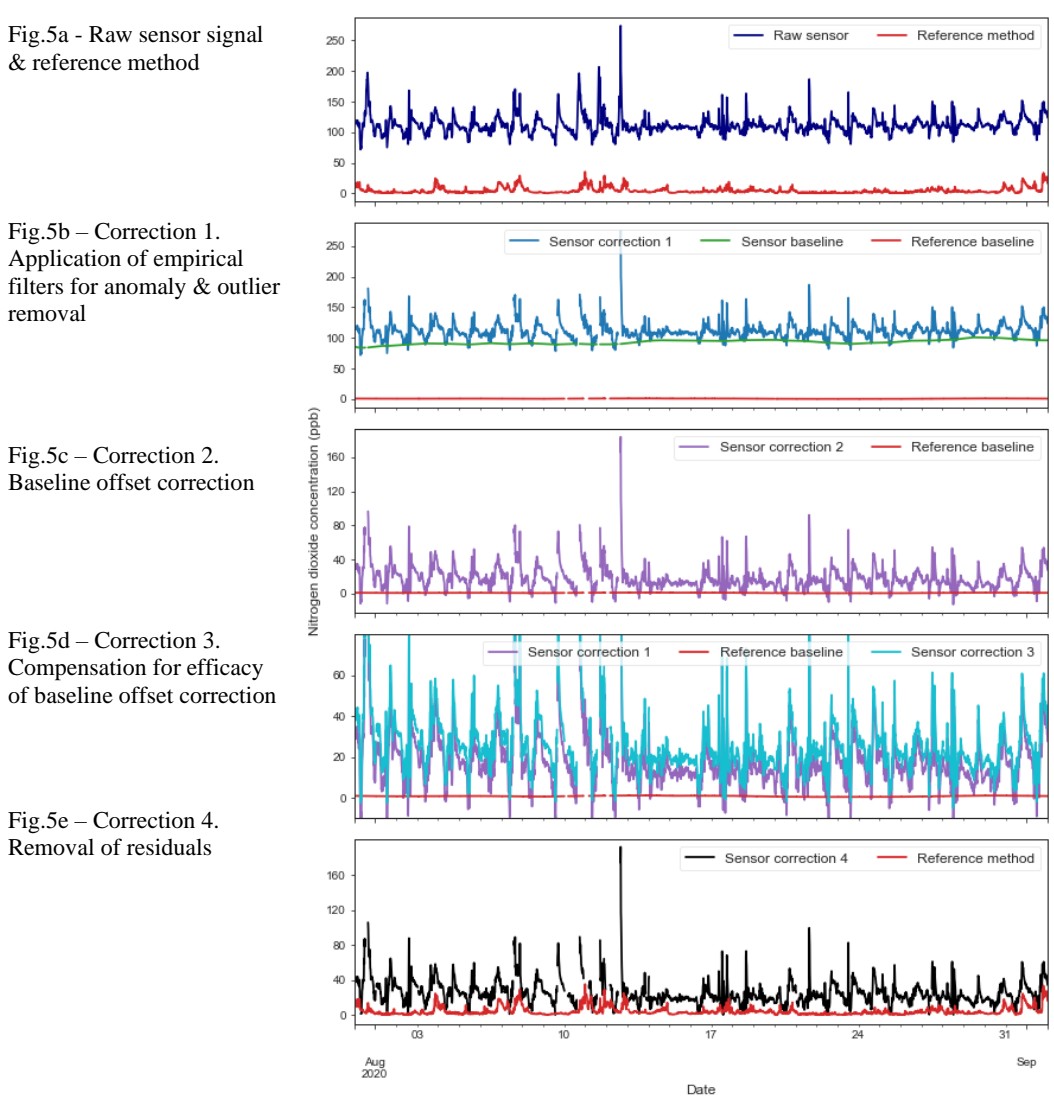





**Figure 6. Relationship between uncorrected, RF model corrected sensor and reference method observations for NO₂, the dotted line shows the unity slope.**

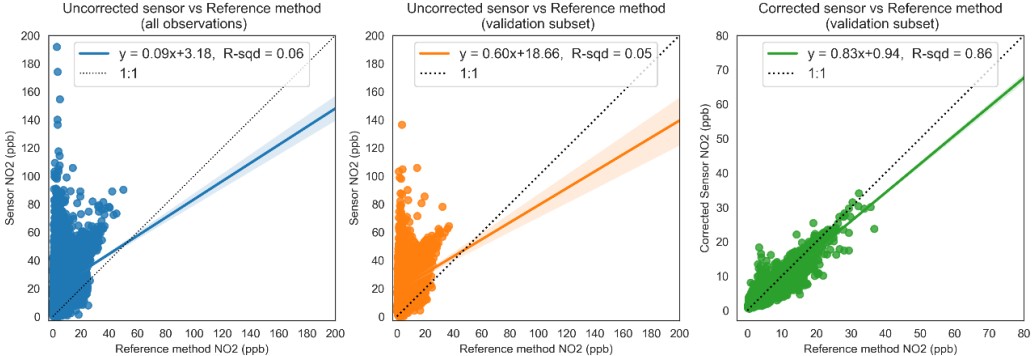

**Figure 7. Relationship between uncorrected, RF model corrected sensor and reference method observations for PM₁₀, the dotted line shows the unity slope.**

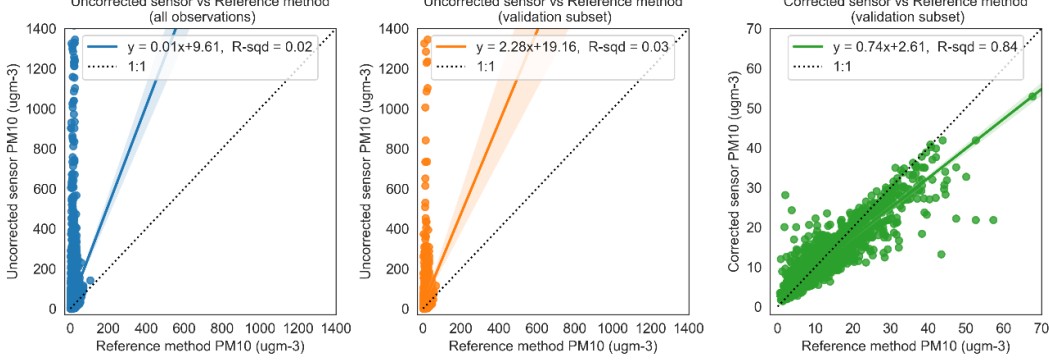


**Figure 8. Relationship between uncorrected, RF model corrected sensor and reference method observations for PM₂.₅, the dotted line shows the unity slope.**

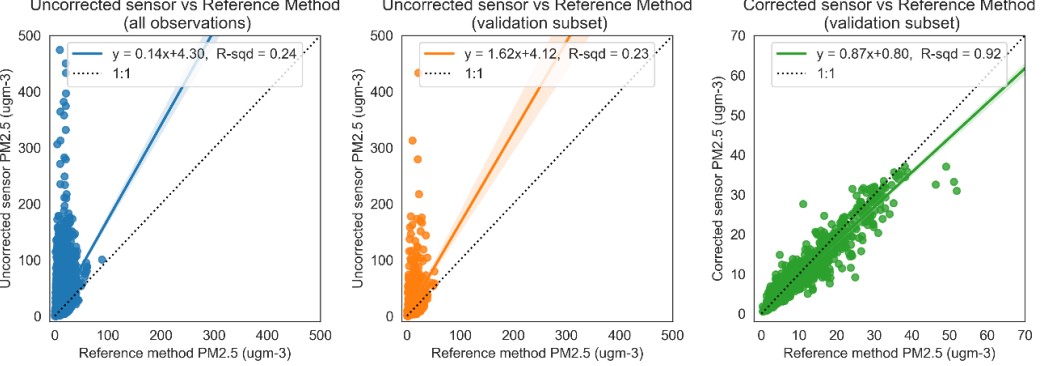





**Table 4. RFR correction model performance in terms of MAE relative to reference method observations, validation data, June to November 2020.**

| | Mean absolute error (MAE) | | Coefficient of determination ($R^2$) | | Change in MAE arising from RFR correction |
|---|---|---|---|---|---|
| | Baseline correction only | Baseline + RFR correction | Baseline correction | Baseline + RFR correction | |
| $NO_2$ (ppb) | 16.8 | 1.2 | 0.05 | 0.86 | 93% |
| $PM_{10}$ (µg/m³) | 33.1 | 1.8 | 0.03 | 0.84 | 95% |
| $PM_{2.5}$ (µg/m³) | 9.0 | 0.9 | 0.23 | 0.92 | 90% |

**Figure 9. Time series of uncorrected-baseline-normalised, fully corrected sensor observations and reference method observations for $NO_2$ St Ebbe's Oxford 2020.**

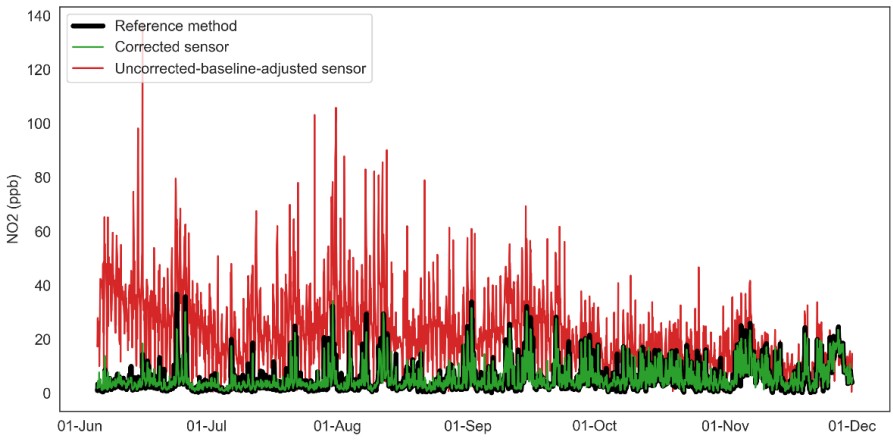





**Figure 10. Time series of uncorrected-baseline-normalised, fully corrected sensor observations and reference method observations for PM₁₀ St Ebbe's Oxford 2020.**

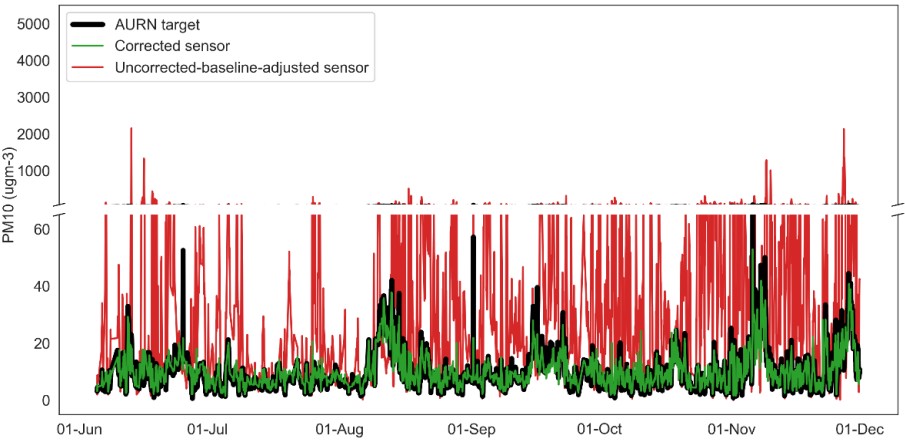

**Figure 11. Time series of uncorrected-baseline-normalised, fully corrected sensor observations and reference method observations for PM₂.₅ St Ebbe's Oxford 2020.**


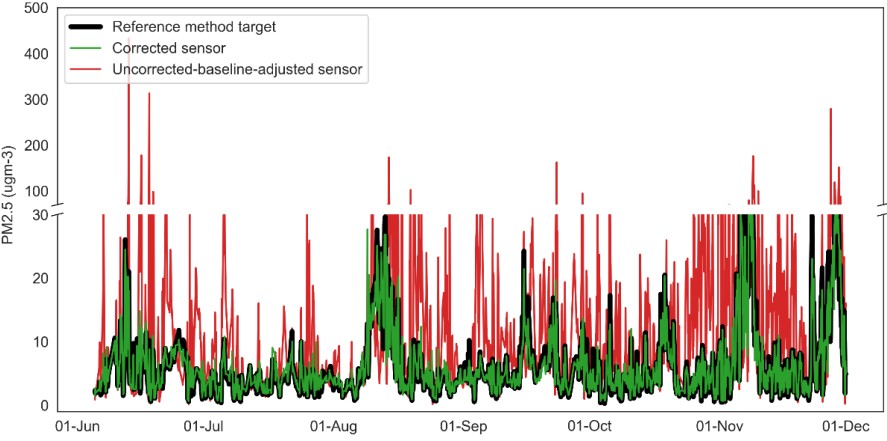

**Table 5. RFR correction model performance in terms of MAE relative to reference method observations, unseen data December 2020.**

|  | Mean absolute error (MAE) | | Change in MAE arising from RFR correction |
|---|---|---|---|
|  | Baseline correction | Baseline + RFR correction |  |
| NO₂ (ppb) | 4.1 | 2.6 | 37% |
| PM₁₀ (µg/m³) | 75.8 | 4.6 | 94% |
| PM₂.₅ (µg/m³) | 10.0 | 2.8 | 72% |



**Figure 12. Time series of uncorrected-baseline-normalised, fully corrected sensor observations and reference method observations for NO₂ St Ebbe's Oxford, unseen data, December 2020.**

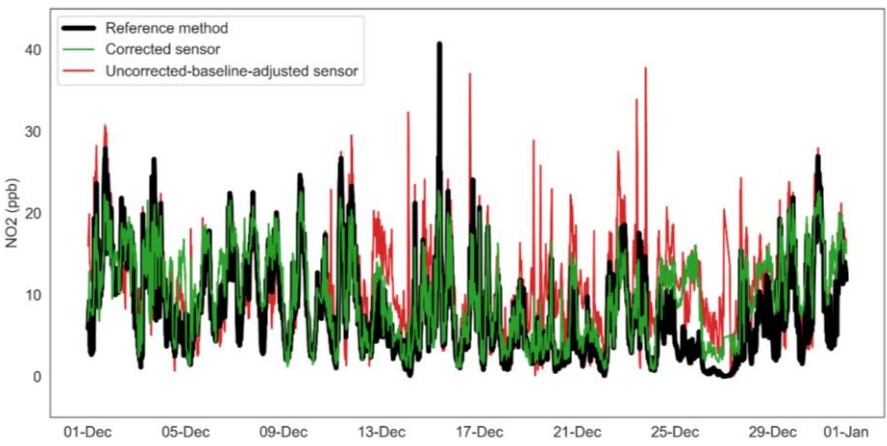

**Figure 13. Time series of uncorrected-baseline-normalised, fully corrected sensor observations and reference method observations for PM₁₀ St Ebbe's Oxford, unseen data, December 2020.**

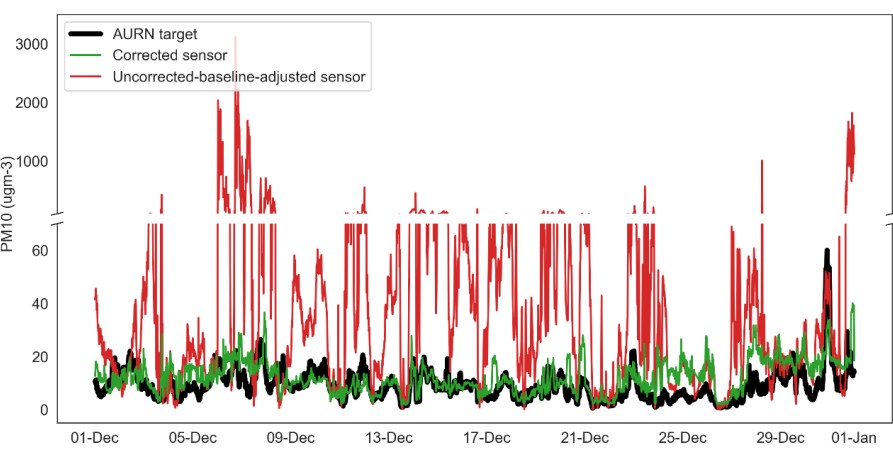






**Figure 14. Time series of uncorrected-baseline-normalised, fully corrected sensor observations and reference method observations for PM$_{2.5}$ St Ebbe's Oxford, unseen data, December 2020.**

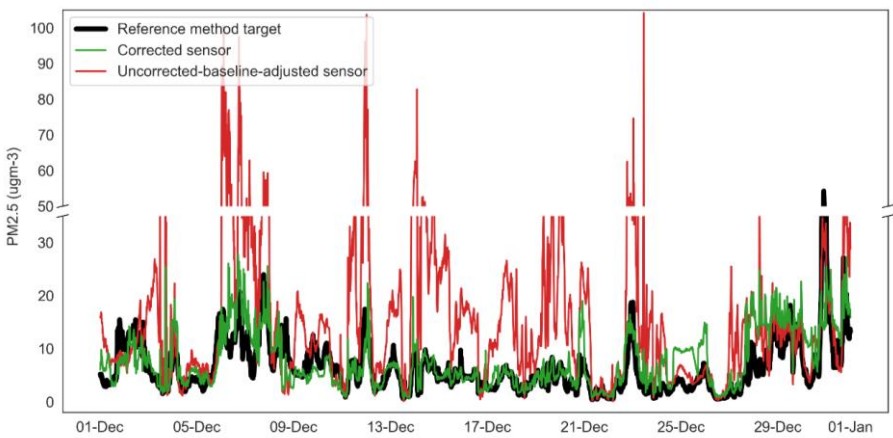


**Table 6. Expanded uncertainty estimates for corrected sensor observations using RFR validation dataset, the target values are the target expanded uncertainty criteria recommended by European legislation.**

| Pollutant | Expanded Uncertainty | Corrected Expanded Uncertainty[a] | R-squared Value | Conformance with Target Expanded Uncertainty Objective |
|---|---|---|---|---|
| NO$_2$ | 21% | 4% | 0.86 | True, ≤25% |
| PM$_{10}$ | 34% | 12% | 0.84 | True, ≤50% |
| PM$_{2.5}$ | 18% | 10% | 0.92 | True, ≤50% |

a. expanded uncertainty estimates with allowance to correct for non-zero intercept and non-unitary slope in the

linear regression relationship of sensor to reference method.

**Table 7. Expanded uncertainty estimates for corrected sensor observations from unseen dataset, December 2020.**

| Pollutant | Expanded Uncertainty | Corrected Expanded Uncertainty[a] | R-squared Value | Conformance with Target Expanded Uncertainty Objective |
|---|---|---|---|---|
| NO$_2$ | 21% | 10% | 0.87 | True, ≤25% |
| PM10 | 34% | 24% | 0.27 | True, ≤50% |
| PM$_{2.5}$ | 29% | 29% | 0.45 | True, ≤50% |

a. expanded uncertainty estimates with allowance to correct for non-zero intercept and non-unitary slope in the

linear regression relationship of sensor to reference method.
