# Peer review of "Machine Learning Techniques to Improve the Field Performance of Low-Cost Air Quality Sensors"

_Atmospheric Measurement Techniques, 2021_

## Author Response (AR1)

**Machine Learning Techniques to Improve the Field Performance of Low-Cost Air Quality Sensors**

**RC comments and AC responses**

RC1: ['Comment on amt-2021-282'](), Anonymous Referee #1, 19 Nov 2021

General Comments:

In this work, the authors developed a machine learning calibration process that combines a 4-stage baseline offset correction and Random Forest Regression Modelling (RF). They adjusted the RF model by identifying readily available training features and optimizing the number of leaf nodes and trees. This work compared the performance of the RF correction model against values from a reference monitor, the raw sensor value, and baseline-corrected sensor values over a time span of ~7 months. This baseline + RF model improved the performance of low-cost $NO_2$, $PM_{10}$, and $PM_{2.5}$ sensors relative to the raw and baseline-corrected values. This machine learning technique is a reasonable method to improve data quality from low-cost air sensors and is suitable for publication after minor revisions.

Major:

RC2-1. Alphasense NO2-A43F electrochemical NO2 sensors (and Alphasense NO2-B43F) have a known cross-sensitivity to ozone (Spinelle et al.). Although the Praxis Urban sensor system and the St Ebbe's monitoring site do not appear to measure ozone, the study fails to mention/address this concern. While inclusion of this variable into feature training could restrict the spread of this model to other networks, it could greatly enhance the performance of the NO2 model. Spinelle et al. also found that sensors from the same manufacturer can behave differently in the same environmental conditions. This manuscript would greatly benefit from applying your model to more than one sensor to demonstrate its capability to nullify discrepancies from sensor to sensor. (Spinelle, L.; Gerboles, M.; Kotsev, A.; Signorini, M. Evaluation of Low-Cost Sensors for Air Pollution Monitoring: Effect of Gaseous Interfering Compounds and Meteorlogical Conditions; Publications Office of the European Union:Luxemborg, 2017. https://doi.org/10.2760/548327)

AC. Firstly, our thanks for your time and thought in preparing your very helpful comments.

Thank you for this comment, we think it is very valuable for context and in developing further learnings. We agree that it is worthwhile adding a note on cross sensitivity with ozone and (will) include a reference to Spinelle 2017 in the revised manuscript

We confirm that ozone data is available at the St Ebbes monitoring station and agree that these data would help in evaluating the effectiveness of ozone as an additional training feature for the development of the RF model and improved correction model performance. However, only 6 in 16 sensors deployed across our network have ozone monitoring capability and this was not the focus for application of low-cost sensor data for local air quality management.

Documenting the performance of the models as-is, is valuable as a demonstrator for the performance that is achievable with the constrained approach presented (i.e. without the ozone cross sensitivity training), not least as this is representative of many real-world low-cost sensor applications where (many) $NO_2$ only electrochemical sensor network in operation.

| AC mods: | Lines 65-75, reference to Spinelle 2017 & commentary. |
| | Lines 225-235, response to O3-NO2 cross sensitivity. |

RC2-2. It is unclear how this model could be applied to sensors throughout a network. Would each sensor need to spend *x* number of months at a reference site to develop the model prior to deployment? How well would a baseline established at the reference site transfer to the deployment site?

AC. For deployment in real world situations I would anticipate that the model, or a variant thereof, would be training for each 'local' network and this model would be directly deployable across a local network e.g. within a town or small city where the influencing variables are likely to be consistent. The correction model itself is constrained by the diversity of data used to train it, both in terms of variability sensor to sensor and in terms of the pollution/environmental conditions to which the sensors are exposed (mainly $NO_2$ & RH). The more diverse the training data, the greater the applicability of the model. One of the main challenges for most applications, and particularly in a study environment such as Oxford which has generally / relatively good air quality, is the under-representation of higher pollution events in the training datasets which may result in over correction (under prediction) of real-world concentrations. In an ideal situation one could imagine co-location at low, medium, high and very high pollution conditions, but as I am sure you are aware such situations are almost impossible to engineer.

| AC mods: | No mods required. |

RC2-3. Line 163: "The filtering criteria presented in Table 1 were identified empirically from an analysis of typical sensor performance from the sensor network and from similar parameters logged at the St Ebbe's AURN station" It is not fully clear how these criteria were chosen. Was this based on limits set by the sensor manufacturer? Please clarify. It would also be useful to state the sample population percentage that was removed based on these criteria, as you did on line 188.

AC. Thank you for this comment, we clarify these criteria were developed independently of the manufacturer. Please see sections 2.3.1 to 2.3.4 for an explanation of the derivation of the filter criteria and associated techniques. We will add a footnote to Table 1 to reflect this.

AC mods:       Line 629, foot note to Table 1.
                      Line 172, revised description of method.
                      Line 181, added proportion of sample population removed.

Minor:

RC2-4. Line 69: "multiple linear regression (MLR) models have been successfully used with variable results" Conflicting statement, please clarify.

AC. We suggest modifying this to "multiple linear regression (MLR) models have been developed with variable results"

AC mods:       Line 68, modified text, as above.

RC2-5. Line 136: Please provide more information regarding the location of the sensor relative to the reference instrumentation.

AC. We confirm that sensor and reference instrumentation were co-located at St Ebbes with sensor inlets were within 0.5 metres (gases) and 2 metres (particles). We will add this to the paper

AC mods:       Lines 146: dimensions added.

RC2-6. Table 4 & Table 5: Please re-format the column headers as it is currently difficult to differentiate between them.

AC. Thank you for this comment, Tables 4 and 5 have been re-formatted.

AC mods:       All tables reformatted.

RC2-7. Line 319: "The performance of each component of the correction method is presented in Table 3" Should read Table 4 I believe. All table references after this point in the manuscript need to be shifted +1 up to Table7.

AC. Thank you for this comment we have corrected the table referencing.

AC mods:       Table numbering reviewed & updated throughout

RC2-8. Line 392: "December 2020 saw the occurrence of several pollution events in the particle sensor time series (as also noted above). Although these events were observed throughout Oxford in multiple particle sensor time series, they were not reciprocated in reference measurements, nor in NO2 data" It seems that around 12/25 in Figs 12-14 all corrected sensor values for NO2, PM10, & PM2.5 experience an increase relative to the reference value. Therefore, it does seem like some event affected all three pollutant models. Have you investigated these anomalies further to locate a common factor?

AC. Yes, we confirm this is correct, NO2, PM10 and PM2.5 sensors were all affected by a series of events in Dec 2020 which were not reciprocated in either PM or NO2 reference data and shown in Figs 12-14. We have undertaken  some further detailed investigation but have no evidence for associated changes in T & RH local sensor time series nor in independent high resolution weather data. I will modify the text to indicate that no evidence was found in the reference datasets for reciprocal events.

AC mods:        Line 388: Commentary on the events provided including to reference to Figs
                illustrating the effects on PM concentrations

RC2: ['Comment on amt-2021-282'](), Anonymous Referee #2, 14 Dec 2021

In this work, the capabilities of low-cost sensors for enhancing urban air quality networks is investigated. Statistical and machine learning methods (Random Forest regression) are used for sensor data post-processing and thus for improving the data quality. It is then evaluated whether the achieved corrected sensor data meets European data quality objectives. It is found that the sensors meet the requirements for "indicative" measurements and it is stated that the sensors are "likely to deliver at least comparable data quality to passive sampler methods (for NO2)". These are important findings that might have impact on regulatory air quality measurements. However, I think that the found conclusions are not sufficiently supported in the way this work is presented. I therefore recommend major revisions before this work can be published. My main comments and concerns are the following:

RC2-1. The applied data post-processing approach is in my view not sufficiently explained. The different applied stages are described, that is good, however, some of the stages raise questions: The filters applied in stage 1 are presented in Table 1. If I understand the logic behind the filters as presented, I conclude that all observations at relative humidity > 35% had to be removed. It is unlikely that this is true, please correct (if yes the sensors are useless for most locations).

AC. Firstly, may I extend our thanks for your time in reviewing the paper and the helpful comments. I can clarify that sensor observations of NO2, PM10 and PM2.5 with associated RH values < 35% were excluded from subsequent analyses. Low relative humidity is generally infrequent in Oxford and the UK because of the maritime climate. Looking at Oxford meteorological records in the last 7 years, there has only been 1-day when RH was <35% as a daily mean. However, RH is likely to vary much more at higher time resolutions than this and in preparing our filters we used 15-minute data for Oxford from an independent source http://eodg.atm.ox.ac.uk/eodg/weather/index.html to reality check our assumptions. These data showed that there were ~1,400 15-minute periods (~2.5 weeks) in 2020 when RH values were <35% in Oxford during 2020. On this basis, though the choice of the 35% RH threshold is a precautionary (conservative) measure to screen out sensed values logged during periods when sensor faults may have occurred, we don't believe this inappropriately biases our data. We have updated text and tables accordingly.

AC mods: Line 176, clarification provided

RC2-2. For stage 2, the authors refer to the original publication (and source code) for information about the applied baseline and drift correction method. Without consulting the original paper, the reader has no information how baseline and drift correction technically has been done. Some brief technical description about the applied method would be helpful and should be provided, maybe also in the form

of supplementary information. If I understand correctly, then stage 2 forces the baseline to be zero and by doing so, sensor drift is also corrected.

AC. Thank you for this observation, we will provide a brief description of the airPLS algorithm for clarity. I can also confirm that your understanding is correct. Stage 2 uses the airPLS technique to correct or normalise sensor offset. Offsets of course do vary from sensor to sensor and over time and the airPLS technique offers significant utility in providing a flexible, fast and programmatically easy way of handling them.

AC mods:        Lines 185-196. Add commentary on airPLS technique

RC2-3.  Stage 3 then compensates for this zeroing and adds an urban scale background concentration. Based on the measurements from an urban background reference site, constant background concentrations have then been determined and added. Firstly, there is no information given how the values for the average uplift have been determined. It is necessary that the authors describe how the given values have been obtained.

AC. The uplift is calculated at the same resolution as the urban background reference i.e. 15-minute resolution. Raw sensor data (at 10s resolution) are aggregated to 15-minute average resolution to align with the same temporal datum as the reference dataset.  The requisite baseline uplift is then calculated by difference for each 15-minute observation. We will add this additional information into the revised manuscript.

AC mods:        Line 202, 339, 361, 369, 375, clarification on the time resolution at which the
                uplift/compensation is calculated.

RC2-4.  Secondly, an urban background concentration that is constant over time appears to be an oversimplification. This assumption should be explained and justified. If this approach is in a real world application applied to a sensor network across a city, then this would also mean that the urban background is assumed to be constant in time and across the entire city. This is ways too simple. The authors themselves state on page 10 that "the availability of a reliable and high-quality city background ... is essential". Please discuss the consequences for bias and error and potential limitations of this oversimplified approach for background determination.

AC. Using the compensation method described above, the uplift is time varying. We agree that its application is limited to the spatial representativeness of the urban background field characterised by the reference location. The reference location used is in this research is part of the UK compliance monitoring network and conforms to stringent siting criteria set by European air quality Directives to promote local respresentivity. In addition, the study area, Oxford, is a relatively small city with uncomplicated local and surrounding topography, and well understood emissions and emission sources. We do not feel, therefore, that the method is over simplified. However, in larger cities and

places with complex terrain, topography or emissions, we agree that over-simplification of the real-world may occur. In such cases it may be prudent to use multiple reference stations to characterise baseline conditions. We will add such caveats to the paper.

AC mods:     Line 169, 487, clarification of the time resolution of compensation method & likely representeviness of the urban background within this study.

RC2-5.  In Figure 5 an example of the processing of raw sensor data from stage 1 to stage 4 is presented for NO2 from the sensor system that was co-located at the reference station. For the final data as shown in Figure 5e, the agreement between corrected sensor data and reference NO2 must be considered as very poor. The sensor data is biased high by about 20ppb and shows a very different temporal variability. The data quality as expressed by the MAE and presented in the result section are certainly not achieved during the shown time period. The authors should explain the shortcoming of their data correction method here.

AC. Thank you very much for this observation. To clarify, Figure 5 only presents, in an illustrative way, the handling of sensor offset (and its possible drift over time). This is a preparatory step prior to correcting sensor interference effects using the RF regression model also described in the paper. We agree that the agreement between corrected sensor data and reference NO2 in Fig 5e overall is poor. However, agreement in the baseline of the corrected sensor and reference method datasets is good. These part-corrected (baseline corrected) data are passed to the RF model to correct for environmental interferences. We include a paragraph at the end of section 1.2 and start of section 2.3 to this effect. We will add an explanatory footnote to Figure 5 for clarity.

AC mods:     Line 650, footnote added to Figure 5.
             Lines 163, 210, clarifications added to clearly demarcate sensor offset correction model & environmental interference models

RC2-6.  The authors write in the methods and materials section (section 2.2) that 16 sensor units were deployed across the city of Oxford. One of the sensor units was co-located at the St. Ebbe's reference station. Most results of this research has been obtained from the co-located sensor unit (albeit sometimes not explicitly stated), only data from two of the remaining 15 sensors has been used for this study (for Figure 4). I find mentioning the sensor network somewhat misleading, when in fact most of the data is not used. But more importantly, there is no information provided about how the sensor units have been calibrated before deployment. The only information about calibration is given in section 3.1, however, it remains unclear if the sensor units were deployed after factory calibration or the authors performed a lab calibration. This should be explained in more detail. Then, I wonder about the huge (up to 80ppb) and different offsets of the different sensor units as shown in Figure 4. How can this be explained when presumable all sensors were

calibrated in the same way? The authors mention these huge and different offsets but do not question them. I think the authors should discuss these offsets and provide an explanation. As an user, I would be alerted when seeing such a behaviour of calibrated measurement systems.

AC. Thank you for this observation we have deleted reference to the 16 sensors in section 2.2 to avoid confusion.

You are correct also, we do not mention sensor calibration extensively. For information the sensor systems were calibrated by the manufacturers. No other calibration, other than acceptance tests upon receipt of the sensor systems was conducted. We agree the offsets observed are unexplained, for the reasons you allude to. However, in our experience this is not atypical sensor behaviour. It is in our view consistent with real-word sensor data uncertainty that needs to be handled and can be done so with the methods we present. The evidence we present indicates that the methods perform well under the conditions set out. We will add a comment to this effect into the paper.

AC mods:       Line 149 reference to 16 sensors removed.
                     Line 134, now confirms the calibration status of sensors

RC2-7. The main result of sensor performance is the MAE from the unseen data relative to the reference. The numbers in the abstract do not agree with the numbers in Table 5, please correct. The time resolution of the data used for calculating the MAE's should be given.

AC. Thank you very much for this comment, you are correct, we do have a consistency. I have updated throughout.

AC mods:       Updated throughout

RC2-8. In section 3.2.3 the performance of the sensors is compared against European data quality objectives and used the approach as defined for demonstrating equivalence to reference methods. The authors do this for the validation data set and the so-called unseen data. I think the validation data set cannot be used for this purpose. Although the validation data has not been used for model training, it is a random sample of the training data and must be considered as being part of the training data. The uncertainty estimated using the validation dataset (Table 6) are too optimistic. For the unseen data set it can be seen that the performance of the PM sensor is much lower compared to the validation data. The author argue for some very special environmental conditions during the considered time period (December 2020). However, this is probably more a realistic scenario for a real world application and when sensors are used at conditions that deviate from conditions during the model training period. In Table 7 the R2 values for PM10 and

PM2.5 are 0.27 and 0.45 respectively, it is hard for me to believe that this is sufficient for fulfilling the expanded uncertainty objective.

AC. You raise several very important and interesting points here. We feel it is valid to present performance and uncertainty estimates for both the validation set and the out-of-sample (unseen) set. Not least because the differences in the two are not well documented in a peer reviewed setting and we see transparency benefits in doing so and they are in concentration units at least relatively small. Also, I believe the validity of the validation set results depends upon how the methods presented in the paper are applied in an operational setting. A 'traditional' view on the type of correction methods presented might be as a tool that is developed / configured once (or irregularly) and is valid for application many times on (multiple) sensor datasets of the same type. The assumption here being that it is relatively easy to train the model to a sufficiently steady state to deliver satisfactory performance. In such a case, I agree the unseen dataset performance is more relevant. An alternative view is that it is not at all easy to train the model to a sufficiently steady state to deliver satisfactory performance - likely linked to RFs inability extrapolate outside of its training range. Hence, to get to the steady state some very diverse AQ data are needed for training, which of course takes time and resource to acquire. However, until such a time as the data is acquired to achieve steady state, if the model is regularly retrained as new data become available the validation set performance is more applicable. By presenting both, we believe we can allow the reader to make a judgement on which is the most useful for their application and, therefore, the likely uncertainty .

AC mods:          Line 390, further justification / confirmation provided.

Other comments:

RC2-9.  The mean absolute error (MAE) is used in the paper for quantification of the sensor performance. Would be nice to have the formula available to see how exactly this quantity was calculated (could be given as a supplementary information).

AC. We can provide a reference (also below), for MAE (RF modelling in general) in the revised manuscript. https://scikit-learn.org/stable/modules/model_evaluation.html#mean-absolute-error

AC mods:               Line 248, 280, reference(s) provided.

RC2-10.        Random forest regression: My impression is that the hyperparameter settings for training the models allowed very and probably too large trees. In particular the minimum number of samples per node (set to a min of 2 samples per node) appears to be very small and might be prone to overfitting. Please comment this.

AC. The RFR models do not appear overly sensitive to tree size and we have found them to be resistant to overfit. With regard to your min sample split query, our tests showed that this default setting worked well; see the fig below for info. The relatively shallow and uniform gradient of the cross validation &

validation curves suggest over-fit at low min sample split values is not a particular issue. I have added comment to the text to indicate that we have performed a reality-check on the hyperparameters chosen to assess the impact of deviations from the parameters identified.

[Figure]

AC mods:                    Line 279, clarification on reanalysis work to check assumptions used, added

RC2-11.        In section 3.2.2. it is referred to Table 3 but this should be Table 4. The different correction steps are difficult to interpret. Please improve formatting. The wrong numbering of tables also continues for the next tables 5, 6 and 7.

AC. Thank you for this observation, also spotted by another reviewer and amended in the revised manuscript.

AC mods:                    Table numbering reviewed & updated throughout

RC2-12.        Section 3.2.2 the MAE values for corrected NO2, PM10 and PM2.5 are given. The temporal resolution of the data used for calculating the given numbers should be mentioned.

AC. Thank you, for highlighting this oversight. We (will) provide the relevant temporal resolution in the revised manuscript.

AC mods:                    Reviewed & updated throughout

---

## Author Response (AR2)

**Suggestions for revision or reasons for rejection**
**Reviewer 2 (RC2).**

*AC. Firstly, our thanks once again for your time in preparing these detailed thoughts comments.*

1. I find the terminology used for "corrected" sensor data confusing and would prefer to have a terminology that is consistently used throughout the paper (would make reading easier). In Figure 5, the different stages of the applied baseline correction lead to "corrected sensor data". In the following RF modelling step (in particular Fig 6) the baseline corrected data is now considered as "uncorrected data" and the RF step again leads to "corrected data". The authors realise this issue in Figs. 9-11 where the terminology "uncorrected-baseline-normalised" and "fully corrected" is used. Please resolve this and used something like "baseline corrected" and "final (fully) corrected".

*AC. Agreed, we have revised the manuscript using the terms below*

> *Full and final corrected (lines 20, 404, 704, 709),*
> *fully corrected (lines 323, 335, 350, 355, 376, 383, 385, 400, 401, 449, 454, 465,468,469, 472, 668, 670, 676, 679, 687, 688, 694, 696, 702, 708),*
> *uncorrected (line 355, 653,656, 660, 670, 676, 679, 687. 694, 697),*
> *part-corrected (lines 298, 309),*
> *baseline corrected (line 667, 687),*

*AC. In updating figures to accommodate the vocabulary above we have noticed discrepancies in some of the statistics presented. We have updated the manuscript throughout in the interest of transparency & reproducibility. Core conclusions are unchanged.*

2. Section 3.2.4 Sensor performance vs. European air quality data objective. I'm still convinced that the validation data set cannot and should not be used for determination of measurement uncertainty. The validation set is an integral part of the RF model building process. It is a random sample of the data used for model training and should only be used for deciding on the RF model parameters and for comparison of different models or modelling approaches. Performance assessments based on the validation should not be considered as representative for the performance that can be achieved/expected in independent (real-world) measurements. The values are too optimistic. The authors should make this more clear and present and discuss the results/sensor performance based on the numbers in Table 7 (the MAE's from Table 7 are mentioned in the abstract, which is good and correct!)

I think it is generally fine to leave Table 6 in the paper as it provides information on the effect of the performed data correction method. Corresponding numbers are presented in many other papers and they are useful for comparison. However, the authors should make clear that these numbers should not be interpreted as the uncertainties to be expected in subsequent atmospheric measurements as they are too optimistic and not representative for sensor applications. Indeed, I think the comparison of numbers in Tables 6 and 7 is useful for readers who intend using sensors for atmospheric measurements. The authors should make clear that the special situations during the collection of unseen data in December is not an exception but the rule, when using sensors in a real-world setting and therefore should be included in the performance assessment.

*AC. Thank you for this comment. We have revised the manuscript to clearly state uncertainties in validation estimates*

Other comments:

Page 5, section 2.2, fourth line. Typo, should be "were" instead of "work".

*AC. Corrected, thank you.*

In section 3.2.3, page 13 the authors now have a sentence "… within about 1ppb (NO2) and 2-3ug/m3 (PM) of the MAE returned by the model validation set.". Where do these numbers come from? – Prob-ably from Table 4, but the reader does not know as numbers for PM are in Table 4 different. Same in the Conclusions section.

*AC. Thank you. Revised with the inclusion of 1 decimal place & cross-reference to source Table 6 for clarity.*

Page 11, second paragraph, first line, "Fig. ", number is missing.

*AC. Page 11, line 1 references "Figs. 6-8" the version I am editing. Perhaps a pdf issue. We will review to ensure the issue does not perpetuate.*

Figure 5. Should be indicated what is shown here. Also rolling 3h means as in Fig. 4, 15-minute mean values, or different?

*AC. Thank you. We have revised the title of Fig 5 to indicate 15-minute mean averaging period of the data shown.*

---

## Author Response (AR3)

Dear Dr Herckes & Editorial Support

Thank you for your feedback (dated 22 Mar 2022) on the latest draft of our manuscript (AMT-2021-282).

Firstly, may I extend my apologies for the extra resource this issue has required. I hope the additional context below and commentary attached helps to move it along in a positive way.

**First identifying the discrepancies**

We first found discrepancies in the manuscript whilst responding to reviewer#2 comments posted on 4 Mar 2022. The discrepancies arose as a direct result of re-running code to reproduce colour-blind accessible figures for the manuscript. After they were identified, we requested to extend the deadline for responding to reviewer #2's comments, as the MS records will show. This allowed us time to fully consider the implications of the discrepancies prior to resubmission.

**Diagnosing the issue**

Despite our code base being segregated from other research code base(s) to protect it from inadvertent changes, it is not part of a formal subversion system. We have identified that a proportion of the random forest (RF) model code was changed in error resulting in the training parameters for the RF models being overwritten. The timing of the overwrite appears to have been late on in the code base development – sometime after our preferred RF model configurations had already been agreed and documented, but before the results were compiled. As a result, the training parameters presented in Table 3 are correct, but do not deliver the results presented in Tables 4, 5, 6, & 7 in a reproducible way. We believe the change occurred during the preparation of follow-up research - the timing and type of changes reflect this follow-on work, and we anticipate that that code base was copied for re-use / development on the downstream project. After copying of the code to a new development environment for modifications, we believe that changes were subsequently saved back to the original location in error.

**Safeguarding for the future**

We will use code repositories e.g. GitHub, to further protect against similar mistakes in the future. At the time of this current analysis we did not have this procedure in place.

**Progressing the manuscript**

We have since reconstructed the RF models according to the parameters set out in Table 3 and achieved reproducible results. These are now reflected in the most recent version of the manuscript uploaded on to AMT on 21 Mar 2022 and available in track-changes at
https://editor.copernicus.org/index.php?_mdl=msover_md&_jrl=400&_lcm=oc3lcm4w&_acm=get_file&_ms=97838&id=1905110&salt=6825354511725275278.

**General comments on changes identified**

We have carefully considered the implications of the changes prior to re-submitting the manuscript. A point-by-point review of the changes required, is provided in the attached

document. From our review we consider that; (i) there are several groups of small changes mainly restricted to the MAE / R-squared values, which do not change the message(s) conveyed by the study; (ii) there are larger changes in the expanded uncertainty estimates, including one in particular which has resulted in a 20% swing in the expanded uncertainty estimate for the $NO_2$ correction model, based on the unseen data (not used RF model training & validation). We note too, that this latter change, results in this model exceeding the data quality criteria used in Europe to identify suitable methods for 'supplementary assessment' by 5%, where supplementary techniques are defined as techniques used to impart additional spatial context to high-quality reference measurements taken at a single location e.g. Palmes type diffusion tubes etc.

As a result of the review, and despite the changes, we consider that the key messages of our research remain unchanged. Our reasoning is set out below;

1. A relatively simple, effective, and flexible method for improving the quality of AQ sensor data is presented and demonstrated
2. We present evidence on the scale of improvements that the research has achieved and what others might expect, broadly >90% reduction in MAE
3. Despite the model for $NO_2$ not achieving the data quality objectives (by 5%), the scale of improvement uncorrected vs corrected is significant and worth sharing with the AQ and sensor communities
4. Our intention in using expanded uncertainty and associated European criteria / thresholds, was to provide real-world context on the efficacy of the models developed
5. We do not imply and have refrained from recommending a particular sensor or correction method that can / can't, should / should not be used. The correction models we present will require retraining for each application, and it is expected that there will be a small variation in results achieved because of this. However, our evidence suggests that significant improvements can be achieved which may approach or exceed the European criteria. We feel this is valuable to make this research finding available by publication.
6. Co-author, Brian Stacey (BS), is the convenor of CEN TC 264 WG15 (Measurement of $PM_{10}$ and $PM_{2.5}$). This group is responsible for constructing the method for expanded uncertainty calculation and the spreadsheet tool used in this study. BS has indicated that there are legitimate statistical reasons set out by the working group that could be used to improve the expanded uncertainty of the $NO_2$ correction model. These relate to the application of a random error term of ~2.85 in the calculations as they currently stand. If this term were to be removed (which could be justified), the expanded uncertainty estimate for the $NO_2$ correction model would reduce to ~10% which meets the target expanded uncertainty.

**Our position**

Based on this reasoning, our team feels that the changes do not alter sufficiently the overall message of the paper. We have not re-worked the uncertainty calculations (6) because of the extra work involved and again this does not significantly change our findings. We are, however, confident that the research presented as-is, is of high quality,

reproducible, transparent and will be useful to readers in developing a solution to sensor data quality for their own applications.

Also note that, the studies code base and data will be made publicly available, a condition of our Natural Environment Research Council funding [NE/V010360/1]. As a result, traceability and repeatability of code is of utmost importance to the study investigators.

Yours sincerely,

Dr Tony Bush,
Department of Engineering Science, University of Oxford & Apertum (Co-Investigator)

Dr Felix Leach,
Department of Engineering Science, University of Oxford (Co-Principal Investigator and Corresponding Author)

Dr Suzanne Bartington
Institute of Applied Health Research University of Birmingham (Co-Principal Investigator)

**Comments on changes to the manuscript text**

These comments apply to modifications in the associate Tables.

Line 19

We demonstrate improvements of between 37% and 94% in the mean absolute error term of fully corrected sensor datasets; equivalent to performance within ±2.6 ppb of the reference method for $NO_2$, ±4.4 µg/m3 for $PM_{10}$ and ±2.7 µg/m3 for $PM_{2.5}$. Expanded uncertainty estimates for $PM_{10}$ and $PM_{2.5}$ correction models are shown to meet performance criteria recommended by European air quality legislation, whilst that of the $NO_2$ correction model was found to be narrowly (~5%) outside of its acceptance envelope. Expanded uncertainty estimates for corrected sensor datasets not used in model training were 29%, 21% and 27% for $NO_2$, $PM_{10}$ and $PM_{2.5}$ respectively. ~~A mean absolute error of 2.6 ppb, 5.1 µg/m3 and 2.9 µg/m3 for NO2, PM10, and PM2.5 respectively, was achieved for the full and final corrected field-deployed sensors compared to a reference method. When used to correct data collected under environmental conditions outside model training, results meet European data quality objectives, albeit with lower accuracy than data from within the trained range.~~

These changes now ensure the text and numbers quoted in the abstract are consistent with the revised content of manuscript and main messages

Line 337

Clearly, the $PM_{2.5}$ model performs excellently in this respect with an R-squared value of 0. 91 and OLS slope and intercept terms approaching unity.

A 5% change in coefficient of determination, we believe this does not change message - with the help of the baseline & RF model correction, the (corrected) sensor observations explain the majority (~90%) of the variability observed in the reference method.

Line 338

The respective R-squared value for both $PM_{10}$ and $NO_2$ RF models (0. 79 and 0.86) also indicate good model performance.

As above, but a 3% reduction in coefficient of determination. The message is unchanged, corrected sensor observations explain the good proportion of the variability observed in the reference method (~80%).

Line 349

In concentration units this equates to fully corrected $NO_2$ sensor observations within approximately ± 1.2 ppb of the reference observation. Similar comparisons for $PM_{10}$ and $PM_{2.5}$ indicate  fully corrected concentrations within ±0.9 µg/m$^3$ ($PM_{10}$) and 1.9 µg/m3 ($PM_{2.5}$)  µg/m$^3$ of the reference method.

Two modifications being seen here.

Firstly, Reviewer#2 queried the use of integer values in the manuscript text, indicating it raised issues with cross referencing to table values. We have re-introduced the values at 1 d.p. to reflect the tables.

Second, we have ~0.1-0.2 (ppb / ugm-3, depending on pollutant) changes in MAE, arising from the model re-runs. From an AQ perspective this is negligible. Manuscript message is unchanged.

Line 361

The data shown are, as expected, less favourable compared with the validation set, returning higher values for the MAE metric, but for air quality context, within 1.4 ppb ($NO_2$) and 2.5 µg/m$^3$ ($PM_{10}$) and 1.8 µg/m$^3$ ($PM_{2.5}$) of the MAE returned by the model validation set (Tables 4 and 5).

As above, we observe the effect of d.p. change & a small model performance change. The difference between the validation & unseen MAE metrics, which is presented here, is consistent with that of the previous version of the manuscript. Message unchanged.

Line 377

Improvements in MAE attributable to the RF model in the range of 37-94% are shown; equivalent to fully corrected observation within, on average approximately ±2.6 ppb of the reference method for $NO_2$, ±4.4 µg/m$^3$ for $PM_{10}$ and ±2.7 µg/m$^3$ for $PM_{2.5}$.

As above, we observe the effect of d.p. change & an small MAE change, which is consistent with that presented at integer level in the previous version of the manuscript. Message unchanged.

Line 382

Section 3.2.4 has been thoroughly recast to respond to reviewer#2 comments.

Line 388

Table 6 presents expanded uncertainty estimates associated with fully corrected sensor data from the validation dataset, (data not used in the RFR model training) and shows that these data for all pollutants perform well against the target expanded uncertainty criteria recommended by European legislation, (expanded uncertainties of 21%, 40% and 19% respectively for $NO_2$, $PM_{10}$ and $PM_{2.5}$).

6% & 1% increase in expanded uncertainty of the validation set for PM10 & PM2.5 respectively, even so the values are within the data quality objective thresholds.

Line 391

The result of this further correction is presented in Table 6 as the 'full and final correction'. Expanded uncertainty estimates for the validation set with full and final corrections applied were 17%, 15% and 12% for $NO_2$, $PM_{10}$ and $PM_{2.5}$ respectively.

2-3% increase in coefficient of determination for PM, NO2 increased by 13% but still within data quality objectives. Messaging unchanged.

Line 401

Table 7 presents these data for fully corrected sensor observations from December 2020. Table 7 shows the expanded uncertainty estimates for fully corrected unseen sensor data of 29%, 21% and 27% respectively for $NO_2$, $PM_{10}$ and $PM_{2.5}$ are returned.

8% increase in expanded uncertainty (NO2) & reduction for PM10 & PM2.5 of 13% & 2% respectively relative to previous version. PM values remain within the data quality objectives. The expanded uncertainty associated with the NO2 correction model we agree is now outside data quality objectives we quote. However, we feel that the message we wish to convey remains the same - the scale of improvement relative to uncorrected sensor is good & the utility of this relatively simple approach to reducing sensor data uncertainty to approximately acceptable levels is of benefit to AQ sensor community. Our position is supported by a regression analysis present as track

changes (in this document only). The 4 additional plot show for context the relationships between corrected sensor & reference methods for validation & unseen data sets under uncorrected & corrected slope & intercept conditions. In these plots all of which fall outside of the target thresholds we see that the relationship is generally really quite good,

We also maintain that with continued training the expanded uncertainty would improve - the electrochemical sensors used for NO2 are very (more) sensitive to T & RH interference than the OPCs used for PM. The ability of the models to cope with variation its model features to

**Relationship between corrected sensor & reference method - validation set, no slope & intercept correction**

[Figure]

**Relationship between corrected sensor & reference method - validation set, with slope & intercept correction**

[Figure]

**Relationship between corrected sensor & reference method – unseen dataset, no slope & intercept correction**

[Figure]

**Relationship between corrected sensor & reference method – unseen dataset, with slope & intercept correction**

[Figure]

Line 402

Further corrections, for slope and intercept terms, had negligible change on these estimates, (30%, 25% and 28% expanded uncertainty respectively for $NO_2$, $PM_{10}$ and $PM_{2.5}$).

As you indicate, increases in expanded uncertainty across the board. Increases are (now) marginal relative to the validation set.